# Estimation of Complex-Trait Prediction Accuracy from the Different Holo-Omics Interaction Models

**DOI:** 10.3390/genes13091580

**Published:** 2022-09-02

**Authors:** Qamar Raza Qadri, Qingbo Zhao, Xueshuang Lai, Zhenyang Zhang, Wei Zhao, Yuchun Pan, Qishan Wang

**Affiliations:** 1School of Agriculture and Biology, Department of Animal Science, Shanghai Jiao Tong University, Shanghai 200240, China; 2Department of Animal Breeding and Reproduction, College of Animal Science, Zhejiang University, Hangzhou 310030, China; 3Hainan Institute, Zhejiang University, Yongyou Industry Park, Yazhou Bay Sci-Tech City, Sanya 572000, China; 4Zhejiang Key Laboratory of Dairy Cattle Genetic Improvement and Milk Quality Research, Hangzhou 310030, China

**Keywords:** holo-omics, model selection, complex trait, prediction accuracy, breeding program, random effect

## Abstract

Statistical models play a significant role in designing competent breeding programs related to complex traits. Recently; the holo-omics framework has been productively utilized in trait prediction; but it contains many complexities. Therefore; it is desirable to establish prediction accuracy while combining the host’s genome and microbiome data. Several methods can be used to combine the two data in the model and study their effectiveness by estimating the prediction accuracy. We validate our holo-omics interaction models with analysis from two publicly available datasets and compare them with genomic and microbiome prediction models. We illustrate that the holo-omics interactive models achieved the highest prediction accuracy in ten out of eleven traits. In particular; the holo-omics interaction matrix estimated using the Hadamard product displayed the highest accuracy in nine out of eleven traits, with the direct holo-omics model and microbiome model showing the highest prediction accuracy in the remaining two traits. We conclude that comparing prediction accuracy in different traits using real data showed important intuitions into the holo-omics architecture of complex traits.

## 1. Introduction

Genomic prediction is used in various fields and has become an essential tool for the animal breeding program [1]. Due to high throughput sequencing (HTS) technologies, the analysis of genetic data from host and microbiome, altogether termed holo-omics, has led to a growing interest in selecting traits in livestock and studying complex phenotypes. Over the past few years, many methodologies have been designed using the hologenome domains to study nutrient acquisition [2], immune response [3], and body development [4]. It has helped to understand the interconnection and coregulation systems that shape host phenotypes and critical functional pathways [5]. The genome-wide SNP data is widely used to construct an SNP-based matrix termed the genomic relationship matrix (GRM) and predict the accuracy of genomic selection [6].

Similarly, the sequencing data from 16S rRNA has been used to construct a microbial composition-based matrix known as the microbial relationship matrix (MRM), and it is based on relative abundance and an operational taxonomic unit (OTU) [7]. The MRM and cluster analysis of large-scale microbiome data has allowed for research and can predict complex traits, mainly related to feed efficiency and methane emission in animals [8,9,10]. Additionally, microbiome diversity has been linked to various human traits with a moderate prediction accuracy [11] and a higher prediction accuracy than genomic prediction in animals [12]. The GRM and MRM have been used collectively in the holo-omics framework for selection programs in livestock [13]. A significant drawback with the holo-omics framework is that it requires complex statistical models to compute the genetic and microbiome effects, resulting in excessive computational processing and timeframe. The complexity of the model makes it challenging to capture the genetic and microbiome effects jointly. Linear mixed models (LMMs) are key tools to fit SNP and OTUs as random effects to explore complex traits’ genetic architecture [14]. One similar study used the holo-omics dataset in a standard linear model to predict complex traits [15]. 

Inspired by the complicated interaction between the hologenome affecting the complex traits prediction, this paper evaluates the accuracy of three holo-omics interaction models based on (i) covariance between random effects genome-based restricted maximum likelihood (CORE-GREML); (ii) Hadamard product; and (iii) direct genomic and microbiome effect. The above interaction models were evaluated using real animal genotype and microbiome datasets against the noninteractive models that directly use only genomic and microbiome’s solo effects. 

## 2. Methods and Methodology

### 2.1. Real Data Analysis

We have used two publicly available datasets to estimate the prediction accuracy of the complex traits. The first dataset was from the Ruminomics – 1000 cows’ study [16], large enough to perform prediction accuracy, whereas the second dataset was from the gut microbial composition study, which consisted of 207 pigs [12]. From Ruminomics – 1000 cows’ data, we only selected the high-density array genotypes of Holstein cows from UK and Italian farms (*n* = 795). The cows were group-housed in loose housing barns with postpartum between 10 and 40 weeks, had an average live weight of 680 kg (Italy) and 678 kg (UK), received the standard diet for at least 14 days, and had no health issues present in them. In total, 120,321 SNPs were retained based on a minimum allele frequency of 0.01. Prokaryotes’ 16S rRNA short-read sequencing data were downloaded from Short Reads Archive (SRA) under project accession number PRJNA517480. The sequences were analyzed using QIIME2 to generate relative abundance and OTU tables [17]. The total abundance rate at 20% retained 734 OTUs. In addition, we analyzed traits from cattle data that reflect a significant interplay between genome and microbiome, such as milk yield (milk, protein, fat, lactose, and fat-corrected milk (FCM)) and methane emission (CH_4_ g/d, CH_4_ DMI, and CH_4_ ECM). Similarly, from the pig data, we analyzed feed conversion (FC), feed intake (FI), and daily gain (DG), which are related to the gut microbial composition. The pigs belonged to the German Piétrain breed (housed at one experimental farm), with an initial weight of 30 kg and a final weight of 105 kg. The average age of the animals was 188 (±14) days during the slaughter time. The details of the datasets are shown in Table 1. 

### 2.2. Evaluation of the Models

A linear mixed modeling approach was used to perform statistical analysis using the BGLR R package [18]. We have used the Bayesian model (BayesB) to fit the effects in the model as it allows markers to have different effects and variance, along with better measures of fit and simple trait architecture. In addition, we used a Bayesian reproducing kernel Hilbert space (RKHS) based approach to fit the genomic and microbiome data as random effects in the interaction models, as RKHS will have better data-driven performance due to the large data size [19]. 

### 2.3. Noninteractive Models

In Models 1a and 1b, we only considered the number of casual SNPs influencing the complex traits to test the genomic effect. We called these noninteractive models because neither genome nor microbiome effects are fit to predict complex traits (Figure 1). The fixed effects in the models may vary because the datasets we used were from two different studies. The models are as follows:y=Xβ+Zγ+g+e  (1a)
where **y** is the vector of complex traits (from Ruminomics – 1000 cows’ study, i.e., milk yield and methane emission-related parameters), **β** is a vector of fixed effect (i.e., animal farm), γ is the animal’s dietary effect fitted by using BayesB, **X** and **Z** are the corresponding design matrix, **g** is a vector with random animal genomic effect fitted by using RKHS, and **e** is the residual term with residual variance $\sigma_{e}^{2}$. The distribution of the animal genome effect is **g**
*~ N* (0, **G**$\sigma_{g}^{2}$), with **G** being the GRM of an individual’s SNPs and $\sigma_{g}^{2}$ being the additive genetic variance. The GRM was estimated by using the VanRaden method [20]:$$G=\frac{\left(Z-2Q\right){\left(Z-2Q\right)}^{T}}{\sum_{m}2p_{m}\left(1-p_{m}\right)}$$
where **Z** is the gene content matrix with entries 0,1 or 2 for each SNP and each animal, and matrix **Q** contains the frequency $p_{m}$ of each SNP m.
$$y=X\beta +Z_{SD}SD+Z_{pen}pen+g+e\;\;(1b)$$
where **y** is the vector of complex traits (from the gut microbial composition dataset, i.e., FC, FI, and DG), **β** is a vector of fixed covariables (i.e., age and weight measured at the test station and the slaughter weight), **SD** and **pen** are the vectors with slaughter day and pen effect fitted by using BayesB, **X**, $Z_{SD}$, and $Z_{pen}$ are the corresponding design matrix, and **g** and **e** are the same as Model **1a**. 

In Models 2a and 2b, we only considered the microbiome effect, i.e., the number of OTUs affecting the phenotype. The OTUs in 20% of the samples were used to construct the OTU table and derive the MRM. The models are as follows:y=Xβ+Zγ+m+e  (2a)
where the model parameters are as described in Model **1a** except **m**, which is the random effect of the microbiome fitted by the RKHS method following the multinomial distribution **m**
*~ N* (0, **M**$\sigma_{m}^{2}$), with **M** being the MRM constructed from OTUs and $\sigma_{m}^{2}$ being the microbial variance. The **M** matrix was estimated as: $$M\;=\frac{RR’}{n}$$
where **R** is the matrix with log-transformed prokaryote’s relative abundance from all animals and n is the number of OTUs within animals. The matrix **R** was scaled and recombined into a single matrix before the calculation of **M**.
$$y=X\beta +Z_{SD}SD+Z_{pen}pen+m+e\;\;(2b)\;$$
where the model parameters are the same as described in **1b,** and **m** is the microbiome’s random effect, same as Model **2a**.

### 2.4. Holo-Omics Interactive Models

The three holo-omics interactive models consisted of three different interaction frameworks. In the first interaction, the genomic and microbiome effects were coincidentally fitted into Models **3a** and **3b** to estimate the prediction accuracy. These models assume the direct interaction between the GRM and MRM to predict the complex traits, hence they are called holo-omics direct prediction models. The models are as follows:y=Xβ+Zγ+g+m+e  (3a)
$$y=X\beta +Z_{SD}SD+Z_{pen}pen+g+m+e\;\;(3b)\;$$
where the model parameters are the same as described in **1a** and **1b**, with **g** being the animal genomic effect as described in **1a,** and **m** being the microbiome’s random effect, same as in model **2a**.

The second interaction consists of CORE-GREML [21] that estimates the covariance between two random effects, i.e., genomic and microbiome, by fitting the Cholesky decomposition of the GRM and MRM in an LMM. Therefore, these models are called holo-omics indirect prediction models. The models are as follows:$$y=X\beta +Z\gamma +g+m+g_{c}m_{c}+e\;\;(4a)\;$$
where the model parameters are the same as Model **3a** except $g_{c}m_{c}$, which is the holo-omics interaction matrix constructed as follows:$$g_{c}m_{c}=\sqrt{M}.\sqrt{G}\prime +\left(\sqrt{M}.\sqrt{G}\prime \right)\prime \;\;\left(4b\right)$$
where $\sqrt{M}$ is the Cholesky decomposition of the **M** matrix, which is described in **2a,** and $\sqrt{G}$ is the Cholesky decomposition of the **G** matrix, which is explained in **1a**, and $\sigma_{g_{c}m_{c}}$ is the covariance between the genomic and microbiome effect. We assume no correlation between the residuals and any other random effects.
$$y=X\beta +Z_{SD}SD+Z_{pen}pen+g+m+g_{c}m_{c}+e\;\;(4c)$$
where the model parameters are the same as described in **3b**, with **g** being the genomic effect, **m** being the microbiome effect and $g_{c}m_{c}$ is the holo-omics interaction matrix estimated using the CORE-GREML described in **4b**.

The third interaction consists of Hadamard product between the random effects, i.e., GRM and MRM [22]. The Hadamard product matrix is then fitted in the LMM to estimate the prediction accuracy of complex traits and is also called holo-omics indirect prediction models.
$$y=X\beta +Z\gamma +g+m+g_{h}m_{h}+e\;\;(5a)\;$$
where the model parameters are the same as described in **3a** except $g_{h}m_{h}$, which is the holo-omics interaction matrix, also known as the covariance matrix, which captures the covariance between genome and microbiome. It is estimated as follows:$$g_{h}=∶={\;}_{}{\left(G\right)}_{ij}\;{\left(M\right)}_{ij}\;\;(5b)$$
where **G** is the GRM defined in **1a**, and **M** is the MRM defined in **2a**. The symbol ∘ is defined as the Hadamard product for the two matrices, *i* = 1,2,3….,*n* indicates the row, and *j* = 1,2,3….,*n* indicates the column. The symbol ∶= means that the left-hand side is defined by the right-hand side, and **G** and **M** are multiplied entry-wise. The element-by-element product of **G** and **M** is:$${\left(G\circ M\right)}_{ij}=\left[\begin{array}{cccc} G_{1,1}M_{1,1} & G_{1,2}M_{1,2} & \cdots & G_{1,n}M_{1,n}\\ G_{2,1}M_{2,1} & G_{2,2}M_{2,2} & \cdots & G_{2,n}M_{2,n}\\ \cdots & \cdots & \cdots & \cdots \\ G_{i,1}M_{i,1} & G_{i,2}M_{i,2} & \cdots & G_{i,j}M_{i,j} \end{array}\right]$$

The $g_{h}m_{h}$ is a square matrix and $\sigma_{g_{h}m_{h}}$ is the covariance between the genome and microbiome.
$$y=X\beta +Z_{SD}SD+Z_{pen}pen+g+m+g_{h}m_{h}+e\;\;(5c)$$
where the model parameters are the same as described in **3b**, with **g** being the genomic effect, **m** being the microbiome effect and $g_{h}m_{h}$ is the holo-omics interaction matrix estimated using the Hadamard product described in **5b**.

### 2.5. Prediction Accuracy Evaluation

We used a foldid vector (*n* = 10) to cross-validate the model’s prediction accuracy [23]. The vector randomly divided our original sample into 10 sub-samples, and each sub-sample was used as the validation set, and the rest of the folds were treated as the training set. We repeated this process ten times to calculate the square of the correlation coefficient (**r²**) and used the average calculation as the criteria for prediction accuracy. Finally, one-way ANOVA with multiple comparisons was used to find the significance (*p*-value) of estimated prediction accuracy between models. We have used the significance value of α = 0.05 to compare our calculated *p*-values and reject the null hypothesis, i.e., mean predicted values are equal. A *p*-value less than the α-value suggests that our calculated prediction accuracy is statistically significant.

## 3. Results

We evaluated the various holo-omics models and tested how the prediction accuracy changed concerning different genomic and microbiome interactions (Figure 2). We not only tested the individual GRM and MRM but also included the sophisticated holo-omics interaction matrix based on the CORE-GREML and Hadamard product. They have been proven to help estimate and detect covariance between random effects using the RKHS method. We calculated the square of the correlation coefficient (**r²**) by comparing prediction accuracy in different traits and showed important intuitions into the holo-omics architecture of complex traits.

### 3.1. Prediction Accuracy of Complex Traits from the Ruminomics—1000 Cow’s Study

The results of noninteractive and holo-omics interactive prediction accuracy using data from the Ruminomics—1000 cows’ study are shown in Table 2. The holo-omics interactive prediction considering the Hadamard interaction framework (rh) showed excellent accuracy in milk yield and methane emission traits. The prediction accuracy of 0.430 ± 0.002 for milk (*p* < 0.0001), 0.388 ± 0.002 for fat (*p* < 0.0001), 0.433 ± 0.002 for protein (*p* < 0.0001), 0.438 ± 0.002 for lactose (*p* < 0.0001), and 0.418 ± 0.002 for FCM (*p* < 0.0001) was estimated from the model. In methane emission-related phenotypes, the prediction accuracy of 0.581 ± 0.001 for CH_4_ g/d (*p* < 0.0001), 0.376 ± 0.002 for CH_4_ DMI (*p* < 0.0001), and 0.436 ± 0.002 for CH_4_ ECM (*p* < 0.0001) was calculated. The holo-omics direct interaction model (rd) also had a prediction accuracy of 0.581 for ± 0.002 CH_4_ g/d (*p* < 0.0001). The genomic prediction model estimated the lowest accuracy among all the models. The summary of one-way ANOVA with multiple comparisons and test statistics is provided in Appendix A.

### 3.2. Prediction Accuracy of Complex Traits from the Gut Microbial Composition Study

The prediction accuracy was peculiar across all models while analyzing the gut microbial composition study data (Table 3). The prediction accuracy of 0.294 ± 0.007 (*p* < 0.0001) was estimated in FC using the holo-omics Hadamard interaction framework. The accuracy of holo-omics direct prediction (0.378 ± 0.008, *p* < 0.0001) and microbial prediction (0.281 ± 0.007, *p* < 0.0001) was highest in DG and FI, respectively. Similar to the previous analysis, the genomic prediction accuracy was lower than the other models.

## 4. Discussion

This study shows a simple approach for modeling holo-omics interactions in LMM to evaluate the prediction accuracy of complex traits in animals. Evaluating models for prediction accuracy before the actual experiment and formulating precise breeding programs using hologenomes in animals is essential. Furthermore, since breeding programs are complicated and expensive, it is necessary to optimize the breeding parameters, which include the size of the population, selection of animals, subject’s age, choosing the phenotypes, and density of genotyping concerning the animal’s age, along with 16s rRNA sequencing. Therefore, statistical models can be effectively used to design and optimize breeding programs with respect to the holo-omics domain. Furthermore, it allows for predicting expected outcomes by experimenting with many alternatives, such as holo-omics interaction frameworks, and proves to be computationally less demanding. One approach is to establish the accuracy of models that can successfully predict complex traits in a breeding program. Additionally, it provides insight into essential factors, such as host genome, epigenome, transcriptome, metagenome, meta-transcriptome, proteomes, metabolome, metaproteome, and meta-metabolome.

The prediction accuracy models evaluated here account for the contribution of the host genome and microbiome to successfully predict complex traits concerning milk yield, methane emissions, and gut microbial composition. Since multiple interactions of holo-omics are involved in shaping these traits, it is essential to show to what extent the models can contribute to a breeding program. Ultimately, this will enable optimization with hologenome data, including the animal’s genotype and phenotype data, for complex traits. In addition, it also helps to understand the role of holo-omics at multiple stages of the selection program.

The models for prediction of the accuracy of holo-omics interactions in animal data were developed on three concepts: (i) estimating the holo-omics interaction matrix by fitting the Cholesky decomposition of the GRM and MRM, termed as CORE-GREML, (ii) estimating the holo-omics interaction matrix by fitting the Hadamard product of the GRM and MRM, and (iii) direct holo-omics interaction of GRM and MRM. Methods to model these three concepts were developed, and the prediction accuracy was calculated using the real animal genotype and microbiome data from two different studies. However, further validation of these three concepts can be possible, provided that the dataset is larger than the current data used in this study to estimate the prediction accuracy of the complex traits.

### 4.1. Modeling the Non-Interactive Interaction

Genomic prediction accuracy has gained attention due to its advantage in formulating breeding programs. Estimating accuracy from genomic prediction models largely depends on the reference population’s size, the heritability of the trait, effective population size, marker density, etc. [24]. Several studies have experimented with these parameters to test which models gain better prediction accuracy and can be further used in breeding programs, considering complex traits. In contrast, microbial flora has also gained importance in shaping complex traits [9,25,26], and estimating prediction accuracy by using metagenomic profiles has shown its precision. Camarinha-Silva et al. [12] studied the gut microbial community’s role in predicting complex traits and showed that the microbial prediction generated a higher accuracy than the genomic prediction. Similarly, in our study, the prediction accuracy also agrees with the previous estimations, i.e., the microbial prediction accuracy (**r_m_**) was higher than the genomic prediction accuracy (**r_g_**) in all the eleven complex traits (Table 2 and Table 3). The gain in prediction accuracy shows the beneficial role of the microbiome in designing breeding programs for complex traits. The term microbiome-association index has been successfully used to explain the fraction of the microbial community involved in analyzing complex traits and reliably estimating the prediction accuracy [27]. In one such study, metagenomic profiles have been successfully used to predict the accuracy of phenotypes in humans and cattle [7]. The considerable prediction accuracy of tested phenotypes, i.e., 0.423 in inflammatory bowel disease (IBD), 0.422 in body mass index (BMI), and 0.553 in methane emission, highlighted the microbiome’s significance and importance in outlining the complex traits.

Machine learning techniques have also shown metagenomics’ effectiveness in random forest and neural network models. The study concluded the better performance of the random forest model in predicting dissolved organic carbon from plant litter decomposition by using microbial community profiles [28]. Although most of the studies estimated the increase in the accuracy of microbial prediction, still one main question needs to be answered whether the microbial prediction accuracy remains the same at different stages of the host’s life since variation in microbe flora is common [29]. Another challenge includes the difference in the microbial community observed due to the host’s ecosystem [30,31]. The above challenges require further research to conclude the reliability of microbial prediction for complex traits, which is outside the scope of our current study.

### 4.2. Modeling the Interactive Interaction

Since the popularity of the concept of hologenome, the interaction between the symbionts and the host has become limelight and widely studied by the scientific community [32,33]. The holo-omics interactive models used in our study have outperformed noninteractive models in predicting high accuracy in 100% complex traits related to milk yield and methane emission (Table 2). The holo-omics interactive model also predicted higher accuracy in gut microbial compositional traits in two of three complex traits (Table 3). Difford et al. [34] studied the holo-omics interactions in dairy cows and their specific roles in association with methane emissions. In another study, Weishaar et al. [35] further explored the domain of hologenome and used it to study its effect on animal feed efficiency traits. Nyholm et al. [36] investigated the composite interactions between the host’s genome and their microbiome by combining the multiomics data. This study showed that integrating holo-omics data will help develop methods that will play a significant role in applied and basic biological research. The diverse prediction accuracy across all models while analyzing the gut microbial composition study data (Table 3) also indicates the importance of integrating different levels of holo-omics data.

In our holo-omics model analysis, the most successful model included the Hadamard interaction framework, in which the element-by-element product of GRM and MRM was estimated to construct a holo-omics interaction matrix. The matrix was further used in the LMM to predict the accuracy of complex traits (Figure 2). Previously, the Hadamard product was efficiently used to study the mechanisms of multimodel learning [37]. Merrick et al. [38] also studied genotype-by-environment (GE) scenarios for genomic prediction accuracy using the Hadamard product and estimated the positive correlation between environments using the variance-covariance matrix of the main effects. In contrast, Pérez-Enciso et al. [15] did not find any improvements in prediction with Hadamard products while using different scenarios of host and microbiome interactions. However, their research showed that the holo-omics prediction accuracy is ~50% better than the microbiome or genomic prediction through simulation and real-time data analysis.

Moreover, we have not estimated prediction accuracy using the Kronecker product framework, which could also be another potential model for testing the prediction accuracy of complex traits. However, several studies suggest that the prediction models using the Hadamard product have continually outperformed the Kronecker product model [38] and consist of a similar covariance structure [39]. Therefore, further studies will be required to conclude the performance of the Kronecker product in the holo-omics framework.

## 5. Conclusions

According to Stewart Ian [40], “Life is a partnership between genes and mathematics.” Similarly, this paper describes several approaches for estimating the holo-omics interaction matrix and utilizing them in LMM for complex trait prediction. The Hadamard product interaction is the most successful holo-omics interaction with respect to complex trait prediction accuracy. The mechanisms that shape the host’s genetics remain unknown. The host’s genome and microbiome generally highlight the critical relationship between them and complex traits. This paper supports the host-microbe relationship and provides strategic results for implementing the holo-omics framework in breeding programs to increase complex trait prediction.

## Figures and Tables

**Figure 1 genes-13-01580-f001:**
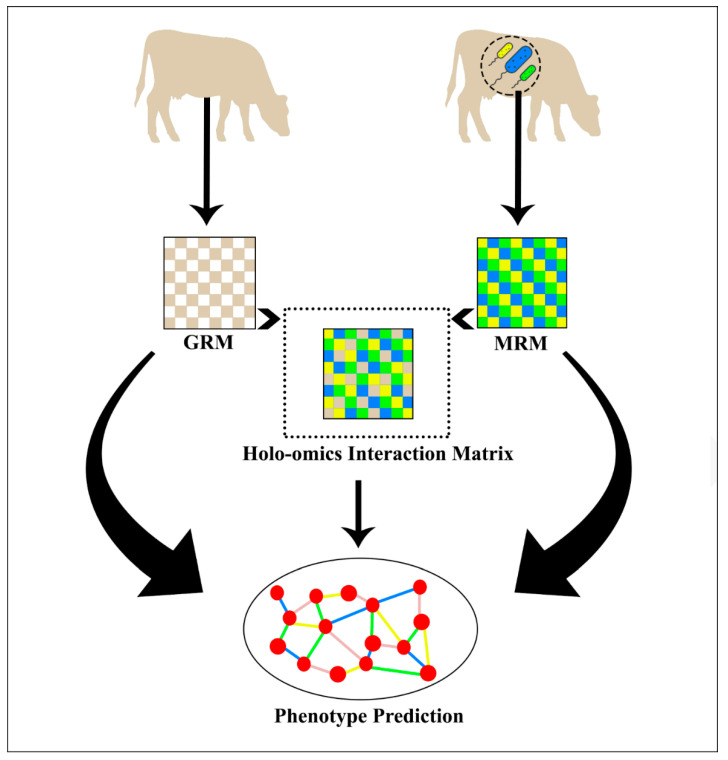
Graphical representation of the models evaluated. The GRM is estimated from an animal’s genome data and the MRM is estimated from the host’s metagenome. The holo-omics interaction matrix is estimated by combining the GRM and MRM in several possible ways. Collectively, all the matrices can be used for phenotype prediction and to study their effects.

**Figure 2 genes-13-01580-f002:**
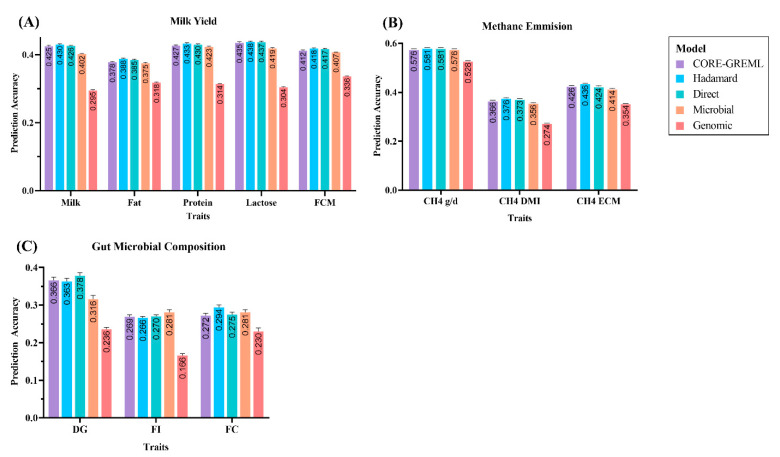
(**A**) Prediction accuracy (**r²**) of different models in milk yield traits (*p* < 0.0001). (**B**) Prediction accuracy (**r²**) of different models in methane emission traits (*p* < 0.0001). (**C**) Prediction accuracy (**r²**) of different models in gut composition traits (*p* < 0.0001).

**Table 1 genes-13-01580-t001:** Details of real animal datasets used to study prediction accuracy.

Name	Subject	Sample size	No. of SNPs	No. of OTUs	Fixed Effects	Traits
**Ruminomics—1000 cows study**	Cattle	795 *	120,321	734	Animal farm	Milk Fat Protein Lactose FCM CH_4_ g/d CH_4_ DMI CH_4_ ECM
**Gut microbial composition study**	Pig	207	51,970	1870	Slaughter weight, weight and age at the test station	Daily Gain Feed Intake Feed Conversion

* From Dataset 1, we filtered only the Holstein dairy cows and used 795 subjects for the analysis.

**Table 2 genes-13-01580-t002:** Accuracy of holo-omics CORE-GREML (r_c_), holo-omics Hadamard (r_h_), holo-omics direct (r_d_), microbiome (r_m_), and genomic prediction (r_g_) of milk yield and methane emission traits. The best estimated prediction accuracy values are highlighted in bold text.

	Holo-omics Indirect Prediction		Holo-omics Direct Prediction	Microbial Prediction	Genomic Prediction
Trait	r_c_	r_h_	r_d_	r_m_	r_g_
Milk	0.425 ± 0.003	**0.430** ± 0.002	0.426 ± 0.002	0.402 ± 0.002	0.295 ± 0.002
Fat	0.378 ± 0.002	**0.388** ± 0.002	0.385 ± 0.002	0.375 ± 0.002	0.318 ± 0.002
Protein	0.427 ± 0.002	**0.433** ± 0.002	0.430 ± 0.002	0.423 ± 0.002	0.314 ± 0.001
Lactose	0.435 ± 0.003	**0.438** ± 0.002	0.437 ± 0.003	0.419 ± 0.002	0.304 ± 0.002
FCM	0.412 ± 0.002	**0.418** ± 0.002	0.417 ± 0.002	0.407 ± 0.001	0.336 ± 0.001
CH_4_ g/d	0.576 ± 0.001	**0.581** ± 0.001	**0.581** ± 0.002	0.576 ± 0.001	0.528 ± 0.002
CH_4_ DMI	0.366 ± 0.002	**0.376** ± 0.002	0.373 ± 0.002	0.356 ± 0.002	0.274 ± 0.001
CH_4_ ECM	0.426 ± 0.002	**0.436** ± 0.002	0.424 ± 0.003	0.414 ± 0.002	0.354 ± 0.001

**Table 3 genes-13-01580-t003:** Accuracy of holo-omics CORE-GREML (r_c_), holo-omics Hadamard (r_h_), holo-omics direct (r_d_), microbiome (r_m_), and genomic prediction (r_g_) of gut microbial compositional traits. The best estimated prediction accuracy values are highlighted in bold text.

	Holo-Omics Indirect Prediction		Holo-Omics Direct Prediction	Microbial Prediction	Genomic Prediction
Trait	r_c_	r_h_	r_d_	r_m_	r_g_
DG	0.366 ± 0.009	0.363 ± 0.009	**0.378** ± 0.008	0.316 ± 0.01	0.236 ± 0.005
FI	0.269 ± 0.005	0.266 ± 0.004	0.270 ± 0.004	**0.281** ± 0.007	0.166 ± 0.005
FC	0.272 ± 0.006	**0.294** ± 0.007	0.275 ± 0.006	0.281 ± 0.007	0.230 ± 0.009

## Data Availability

Publicly available datasets were analyzed in this study. The datasets can be found at https://www.science.org/doi/10.1126/sciadv.aav8391 (Ruminomics—1000 cow's study) and https://www.ncbi.nlm.nih.gov/pmc/articles/PMC5500156/ (gut microbial composition study).

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
