# Peer review of "Estimation of Complex-Trait Prediction Accuracy from the Different Holo-Omics Interaction Models"

_genes, 2022, doi:10.3390/genes13091580_

Round 1

Reviewer 1 Report

As stated by the authors, the manuscript titled “Estimation of complex-trat prediction accuracy from the different holo-omics interaction models” aims to evaluate the accuracy of three holo-omics interaction models on i) COvariance between Random Effects Genome-based restricted maximum likelihood (CORE-GREML); ii) Hadamard product; and iii) direct genomic and microbiome effect.

The manuscript effectively evaluates the accuracy of the different models listed above for predicting complex-traits in cows and pigs. The manuscript also provides convincing evidence that holo-omics indirect prediction with the Hadamard product can improve the prediction accuracy of several complex traits. Below are some suggested changes that could be made to improve the overall quality of the manuscript: 

The in-text references should be changed to the correct format for the journal (numbered references in square brackets).

The ruminomics study needs to be listed in the references at the end (Wallace et al., 2019)

A brief summary should be added to the materials and methods section to describe the study design for each data set used (e.g. animal ages, breeds, housing, etc.). This will help connect parameters in models and the limitations that are addressed by the authors in the discussion.

Include in the description of the models what models are used for the holo-omics direct and indirect prediction to keep consistent with what is presented in the results.

Line 177 – please define the p-value used for significance in text

Line 180 – Figure 2 referenced but results not discussed in text. More information on these results should be provided in the text.

Figure 2 should have significant differences shown. The figure description should state what the numbers represent (r2)

Table 2 and 3 show the best predictive values bolded. Please indicate that in the table description.

It may be worth discussing how the predictive ability of the model could potentially be improved by integrating other metabolomic or proteomic information (e.g. microbial metabolites produced in gut, hormone levels in blood, etc.). This was touched on briefly in the discussion, but could be elaborated on as it may explain some of the results from table 3 in particular. 

Author Response

We would like to thank the reviewers for their useful comments and their positive assessment of our study. Find the attached file which contains the responses to the reviewer’s comments.

Thank you.

Reviewer 2 Report

This an original paper focused on estimation of complex-trait prediction accuracy from the different holo-omics interaction models. Subject of article is current and important because used the holo-omics dataset in standard linear model for predicting complex traits can be useful for the animal breeding program. At the same time, the understanding of biomolecular interactions will allow to optimize and implement this knowledge when applying it to animal production.

Comments:

References are not given in the text according to the instructions for the author.

In Table 1, for Ruminomics it is labeled as a “dietary components” fixed effect, but in models 1(a) it is labeled as an “animal farm” fixed effect – I recommend unifying it.

Author Response

We would like to thank the reviewers for their useful comments and their positive assessment of our study. Find the attached file with responses to the reviewer’s comments. Thanks.
